# Deep Eutectic Solvents as New Extraction Media for Flavonoids in Mung Bean

**DOI:** 10.3390/foods13050777

**Published:** 2024-03-01

**Authors:** Jingyu Gao, Longli Xie, Yu Peng, Mo Li, Jingming Li, Yuanying Ni, Xin Wen

**Affiliations:** 1College of Food Science and Nutritional Engineering, China Agricultural University, Beijing 100083, China; b20223060512@cau.edu.cn (J.G.); xielongli_sicau@163.com (L.X.); yu1.peng@outlook.com (Y.P.); limo0125@cau.edu.cn (M.L.); lijingming@cau.edu.cn (J.L.); nyy@cau.edu.cn (Y.N.); 2National Engineering Research Center for Fruit and Vegetable Processing, Beijing 100083, China; 3Key Laboratory of Fruit and Vegetable Processing, Ministry of Agriculture, Beijing 100083, China

**Keywords:** mung bean, flavonoids, deep eutectic solvent, ultrasound-assisted extraction, antioxidant capacity

## Abstract

Mung beans contain abundant flavonoids like vitexin and isovitexin, which contribute to their strong bioactivities, such as antioxidant effects, so efforts should focus on extracting bioactive flavonoids as well as aligning with the goal of green extraction for specific applications. Deep eutectic solvent coupled with ultrasound-assisted extraction (DES-UAE) was applied to extract flavonoids from mung beans, and eight different DESs were compared on the extraction yield. In addition, the traditional extraction method with 30% ethanol was performed as the reference. The results showed that ethylene glycol-glycolic acid achieved the highest yield among all the DESs, 1.6 times that of the reference values. Furthermore, the DES-UAE parameters were optimized as a 60 mL/g liquid–solid ratio, 30% water content in DES, 200 W ultrasonic power, 67 °C ultrasonic temperature, and 10 min extraction time, leading to the DES extract with the maximum extraction yield of 2339.45 ± 42.98 μg/g, and the significantly stronger DPPH and ABTS radical scavenging ability than the traditional extract. Therefore, employing DES and ultrasonic extraction together offers a green method for extracting flavonoids from mung beans, advancing the development and utilization of plant-derived effective components in a sustainable manner.

## 1. Introduction

Mung bean (*Phaseolus radiatus* L.) has been cultivated in China for many years. It is popular among the public both for food and medicine [1]. Mung beans contain a lot of bioactive compounds, such as protein, carbohydrates, phenolics, and flavonoids [2]. Mung bean showed many potential health benefits because of the existence of bioactive compounds, including hypoglycemic, hypolipidemic, anti-inflammatory, antinociceptive, antioxidative, and antiproliferative effects [2]. As one of the most important bioactive substances, flavonoids gained more attention in recent years [3,4].

Flavonoids have been reported as antioxidants and they are abundant in mung bean, especially in the hull, accounting for 83.9% of the total flavonoids, with vitexin and isovitexin as the main types of flavonoids [5]. Vitexin (Figure 1A) has numerous biological effects, such as antioxidant, anticancer, and antinociceptive effects. Because of their comparable chemical structures, isovitexins (Figure 1B) have biological effects similar to those of vitexins [6]. Researchers have been working to extract mung bean flavonoids because of their excellent functions. Organic solvents like methanol, ethanol, and acetonitrile were typically utilized to extract mung bean flavonoids using the conventional approach [4,7,8]. Furthermore, ultrasound-assisted extraction (UAE) was the method most commonly used to improve the extraction efficiency of mung bean flavonoids because of its high-frequency ultrasonic waves [9]. Huang et al. [7] extracted flavonoids from germinated mung beans using acetonitrile. It has also been reported that flavonoids have DPPH· scavenging capacity (0.13 μmol Trolox equivalent/g germinated mung beans). Natural antioxidants from mung bean (the flavonoids isovitexin and vitexin) extracted using a process that combined ethanol and ultrasonic assistance was studied by Zhou et al. [9]. Although these solvents are relatively effective in flavonoids extraction, they will pollute the environment, and tend to require a long extraction time. As a result, in order to extract flavonoids from mung beans, green solvents are being investigated rather than organic ones, which is generating more and more attention.

Deep eutectic solvent (DES) has received increased attention as an environmentally friendly and long-lasting solvent to take the place of the extraction field’s present harsh organic solvents [10]. DES typically consists of a combination of two or three compounds, which are present in specific proportions. Additionally, DES exhibits a notably lower melting point in comparison to its individual components [11]. Choline chloride (ChCl) is frequently utilized as a hydrogen bond acceptor (HBA), and it can be combined with hydrogen bond donors (HBD) like urea, glycerol, carbohydrate-derived polyols, and carboxylic acids to prepare DES [12]. DES has more promising advantages than conventional solvents, such as non-toxicity, non-flammability, and biodegradability. In addition, it can enhance the solubility of natural compounds [13]. Currently, DES has been utilized to extract bioactive compounds such as phenolic acids from walnut leaves [14], tanshinones from the root of *Salvia miltiorrhiza bunge* [15], and anthraquinones from *Rhei Rhizoma et Radix* [16]. There is currently no available report on the use of DES for extracting mung bean flavonoids.

In this research, our objective was to create an environmentally friendly and effective DES extraction method assisted by ultrasound technology to extract mung bean flavonoids. Five different choline chloride-based DESs and three different alcohol-based DESs were compared firstly on the flavonoid extraction yield. Following this, the optimal DES was selected and tailored for the highest total amounts of flavonoids (TAFs) extraction efficiency by statistical optimization of operational conditions using response surface methodology (RSM), which was assessed based on the extraction yield of the sum of amounts of vitexin and isovitexin. Additionally, we assessed the antioxidant property of DES extract towards 1,1-diphenyl-2-picrylhydrazyl (DPPH) radicals and 2,2′-azino-bis-3-ethylbenzthiazoline-6-sulphonic acid (ABTS) radicals under optimal extraction parameters, comparing to the traditional ethanol extract. At last, to evaluate the interaction mechanism between DES and mung bean flavonoids, Fourier transform infrared (FT-IR) and scanning electron microscopy (SEM) were conducted. The former was utilized to assess how DES and flavonoids interacted within molecules, and the latter was applied to characterize the change in surface morphology of mung beans before and after extraction with various extraction solvents.

## 2. Materials and Methods

### 2.1. Chemicals and Reagents

Choline chloride (ChCl, 98%, CID: 6209), urea (99%, CID: 1176), citric acid (≥99.5%, CID: 311), malic acid (99% CID: 525), ethylene glycol (98%, CID: 174), malonate (99.5%, CID: 9084), glycolic acid (98%, CID: 757), and ethanol (EtOH, 99.9%, CID: 702) were of analytical grade and acquired from Macklin Biochemical Co., Ltd. (Shanghai, China). 1,2-propanediol (AR, ≥99.5%, CID: 1030), vitamin C (VC), DPPH free radical scavenging capacity assay kit, and ABTS free radical scavenging capacity assay kit were purchased from Solarbio Science & Technology Co., Ltd. (Beijing, China). Methanol (CID: 887), acetic acid (CID: 176), water (CID: 962), vitexin (CID: 5280441), and isovitexin (CID: 162350) were of chromatographic grade and acquired from Aladdin (Shanghai, China).

### 2.2. Plant Materials

Dried mung beans were harvested in Baicheng, Jilin, China, in July 2022. All samples were handpicked to ensure no broken beans were used. Then, the whole beans were stored at −80 °C until they were used. The beans were soaked in deionized water for 10 h at a ratio of 1:6 (g/mL) before use. The water was drained off and the beans were used for further experiments.

### 2.3. Preparation of Deep Eutectic Solvents

According to the studies on flavonoid extraction [17,18], hydrophilic DESs were selected to extract mung bean flavonoids. Furthermore, in order to achieve low price, easy preparation, and environmental friendliness, five different choline chloride-based DESs and three different alcohol-based DESs were prepared (shown in Table 1). The mixture of different components was heated to 90 °C by stirring continuously until a uniform liquid formed to prepare DESs.

### 2.4. Ultrasound-Assisted Extraction (UAE) of Total Amounts of Flavonoids from Mung Bean by Deep Eutectic Solvents and EtOH

For initial DES screening, different DESs were prepared with 20% (*v*/*v*) water content. 8.0 mL of extraction solvent was added to 2.0 g of mung bean in a beaker and briefly vortexed. The UAE method was performed at 60 °C and 350 W for 50 min. Following extraction, the mixtures underwent centrifugation at room temperature for 20 min at a speed of 6000 rpm. In total, 1 mL supernatant was taken and diluted with 3 mL water, then filtered through a 0.45 μm cellulose acetate membrane for further HPLC analysis of total flavonoid content. Each extraction was performed in triplicate. To compare the ability of the DES and conventional solvent to extract mung bean flavonoids, the ethanol (30%, *v*/*v*) was also conducted in the same condition.

### 2.5. Single-Factor Experiment

The extraction elements of selected DES were optimized, including the ratio of component 1 and component 2 in DES (6:1, 5:1, 4:1, 3:1, 2:1, 1:1, 1:2, 1:3, 1:4, 1:5, 1:6 mol/mol, as shown in Table 1), water content in DES (0%, 10%, 20%, 30%, 40%, 50%, 60%, 70%, and 80%), liquid–solid ratio (10, 20, 30, 40, 50, 60, 70, 80, 90 mL/g), ultrasonic power (100, 200, 300, 400, 500 W), extraction temperature (30, 40, 50, 60, 70, 80 °C), and extraction time (10, 20, 30, 40, 50, 60 min). The parameters were determined according to our pre-experiment on mung bean flavonoid extraction.

### 2.6. Response Surface Methodology (RSM)

RSM was introduced to explore the optimization and possible interactions of variables, and Design-Expert Ver. 8.0 (Statease Inc., Minneapolis, MN, USA) was used in the process. Using the findings from single-factor experiments, a three-factor Box–Behnken design (BBD) experiment was conducted to identify the most optimal conditions for achieving the highest extraction yield of TAFs from mung beans. This BBD experiment consisted of 17 randomized experiments, with each experiment being replicated three times. As shown in Table 2, three independent variables were selected according to extremes, including liquid–solid ratio (X1, 60–70–80 mL/g), ultrasonic temperature (X2, 60–70–80 °C), and water content of DES (X3, 30–40–50%) at three levels (−1, 0, and +1). Subsequently, three additional confirmation experiments were conducted to confirm the validity of the statistical experimental strategies.

### 2.7. High Performance Liquid Chromatography Analysis

The flavonoids were analyzed on an LC-20 HPLC (Shimadzu Co., Tokyo, Japan) equipped with a diode array detector (DAD) (Shimadzu Co., Japan). A 10 μL sample was loaded on a Venusil ASB C18 reverse phase column (4.6 mm × 150 mm, 5 μm particle size) (Agilent, Santa Clara, CA, USA) through an auto-sampler. A temperature of 35 °C was chosen for the column. Eluent A, which included 1% acetic acid in water, and Eluent B, which contained methanol, made up the mobile phases. The following is the elution gradient: 0–10 min, 10–35% B; 11–25 min, 35–42% B; 26–35 min, 42–75% B; 36–40 min, 75% B; 41–45 min, 75–10% B; 46–50 min, 10% B. The mobile phase was flowing at 1.0 mL/min. The flavonoids could be seen and recognized at 254 nm.

### 2.8. Assessment of Antioxidant Capacity

The antioxidant capacity of flavonoid extracts obtained by DES or 30% EtOH was measured using DPPH· and ABTS· assays. We used the assay kits to determine the scavenging activities of DPPH· and ABTS·, respectively. Vitamin C (VC) was used as a positive control. Each sample was tested in triplicate.

The DPPH· or ABTS· scavenging activity of samples was calculated as follows:$$\mathrm{DPPH}\;\mathrm{or}\;\mathrm{ABTS}\;\mathrm{radical}\;\mathrm{scavenging}\;\mathrm{capacity}\;(\%)=\left(1-\frac{A_{s}-A_{c}}{A_{b}}\right)\times 100$$
where *A_s_*, *A_c_*, and *A_b_* represent the absorbance values at 517 nm for the sample, control, and blank, respectively.

### 2.9. Scanning Electron Microscope (SEM)

We used SEM (Hitachi-S-4800, Tokyo, Japan) to observe the micromorphology of the mung beans before and after extraction by different methods (UAE-DES and UAE-30% EtOH). The samples were fixed onto a copper stub, sputtered with a layer of gold, and then examined by the SEM.

### 2.10. Fourier Transform Infrared Spectrometer

We used a Bruker ALPHA system (Bruker, Bremen, Germany) to record FT-IR spectra of the mung bean flavonoids. Five samples were prepared, including vitexin monomer, isovitexin monomer, DES, vitexin-DES, and isovitexin-DES (prepared by dissolving a specific quantity of vitexin or isovitexin in DES, respectively). FT-IR measurements were carried out in a scan range of 400 to 4000 cm^−1^.

### 2.11. Statistical Analyses

Statistical analyses were conducted using Statistica 6.0 (StatSoft Inc., Tulsa, OK, USA). Data were presented as mean ± standard deviation based on three independent measurements. Significant difference among groups was considered at *p* < 0.05 or 0.01 and analyzed by one-way ANOVA and Dunnett’s *t*-test.

## 3. Results and Discussion

Mung bean flavonoids were extracted by DES and analyzed using HPLC. Reference extraction solvents, such as 30% EtOH, were used, and the UAE was employed as an assisted method because it is a simple and efficient technique [19]. The two primary flavonoids in mung beans, vitexin and isovitexin, were added together to determine the total amounts of flavonoids (TAFs) extracted. As a result, throughout the research, the flavonoid extraction yields were assessed in regard to TAFs.

Firstly, the optimum type of DES for extracting mung bean flavonoids was selected via different solvents assisted by the UAE method. Subsequently, the effects of six independent variables, including the ratio between component 1 and component 2 of DES, the water content in DES, the liquid–solid ratio, the ultrasonic power, the extraction temperature, and the extraction time on flavonoid extraction, were considered. The optimal extraction condition was determined by RSM experiments. Then, the antioxidant abilities to scavenge DPPH radicals and ABTS radicals between DES extract and 30% EtOH extract were compared. Finally, FT-IR and SEM were conducted to evaluate the mechanism of extraction yield improvement by UAE-DES.

### 3.1. Selection of Deep Eutectic Solvents 

Generally, there are many elements that affect the ability of DESs to extract bioactive compounds from plants, including types of HBA and HBD, molar ratio of HBA and HBD, water content in DES, polarity and viscosity of DES et al. [20]. DESs exhibit unique properties, so they have a wide range of applications. It is important to select the suitable type of DES to extract the target compounds from a sample matrix. As shown in Figure 2A, different types of DESs have the ability to extract flavonoids from mung beans, resulting in varying extraction yields. The extraction yields obtained from DES-3 (choline chloride: ethylene glycol = 1:2, mol/mol), DES-6 (ethylene glycol: malonate = 1:2, mol/mol), DES-7 (1,2-propanediol: glycolic acid = 1:2, mol/mol), and DES-8 (ethylene glycol: glycolic acid = 1:2, mol/mol) were 1481.44 ± 12.16, 1466.37 ± 53.09, 1605.40 ± 80.70, and 1985.34 ± 91.67 μg/g, respectively, which were significantly higher than those of 30% EtOH (1234.75 ± 101.56 μg/g, *p* < 0.05). While the attained extraction yields of DES-1 (choline chloride: urea = 1:2, mol/mol), DES-2 (choline chloride: 1,2-propanediol = 1:2, mol/mol), and DES-5 (choline chloride: malic acid = 1:2, mol/mol) were not significantly different from that of 30% EtOH (*p* < 0.05). DES-4 (choline chloride: citric acid = 1:2, mol/mol) had the poorest extraction yield (997.68 ± 9.06 μg/g). To sum up, alcohol-based DESs (DES-6, DES-7, and DES-8) had better extraction efficiency compared to choline chloride (ChCl)-based DESs (DES-1, DES-2, DES-4, and DES-5) and EtOH.

The reasons were the following: On the one hand, the general principle of DES is its ability to donate and absorb hydrogen. This ability allows the formation of hydrogen bonds with the components to be extracted and facilitates the extraction. So, the high extractability of flavonoids with DES may be attributed to the stronger H-bonding interactions between alcohol-based DES molecules and flavonoid compounds [21,22]. On the other hand, since the synthesized DESs have relatively lower polarity, they demonstrate higher extraction yields for low-polar natural compounds, e.g., vitexin and isovitexin, in the present study. Moreover, compared with conventional solvents, the high viscosity of DESs restricts their application as extraction solvents. Our results demonstrate that the less viscous alcohol-based DESs show higher efficiency for naturally occurring compounds. The higher viscosity of ChCl-based DES may hamper mass transfer and diffusion of the target compounds, leading to lower extraction yields of mung bean flavonoids [16,23]. DES-8 exhibited the highest extraction efficiency over the other DESs and 30% EtOH, which was chosen for the subsequent experimentation. A similar result was also found in Wei et al.’s research, in which they concluded that, compared with the traditional organic solvent, the flavonoid extraction ability of the natural deep eutectic solvent prepared by 1,4-butanediol-lactic acid is also significantly improved [24]. It indicated that DESs prepared by short-chain alcohols and organic acids can be employed as green and efficient extraction media for obtaining flavonoids.

### 3.2. Effect of Hydrogen Bond Acceptors-Hydrogen Bond Donors Molar Ratios on the Extraction Yield of Total Amounts of Flavonoids 

The physicochemical properties of DES can be influenced by the mole ratio of HBA to HBD [25]. Therefore, the impact of different HBA-to-HBD molar ratios of DES-8 (ethylene glycol: glycolic acid) on the extraction yields of TAFs was examined. Different molar ratios of DES-8 (ranging from 6:1 to 1:6) were used to prepare the samples, and the outcomes are depicted in Figure 2B. The extraction yield of TAFs fluctuated as the molar ratio between ethylene glycol and glycolic acid changed. The maximum extraction yield (1052.51 ± 26.85 μg/g) was obtained at a molar ratio of 3:1, which exhibited a significant difference (*p* < 0.05) over other parameters. So, the molar ratio of 3:1 was chosen for further design.

### 3.3. Effect of Water Content in Deep Eutectic Solvent on the Extraction Yield of Total Amounts of Flavonoids 

The water content in DES greatly affects its viscosity, making it a crucial factor [26]. Therefore, different water contents of DES-8 were adjusted to optimize the extraction of TAFs from mung beans. The results are shown in Figure 2C. Increasing water content in DES from 0% to 40% presented a gradual increase trend on the TAFs extraction yield from 787.14 ± 59.29 μg/g to 1099.60 ± 36.19 μg/g, while the yield decreased significantly to 810.18 ± 65.25 μg/g with further increase in water content to 80%. The higher extraction yields of TAFs at lower water content may be attributed to the reduced viscosity of DES when the water content is at an optimal level. The reduced viscosity could then potentially enhance the extraction process of flavonoids. However, as the water content increased, the extraction efficiency shrank. The possible reasons were as follows. Too much water even completely disrupts the interactions between DES components, reduces the interaction between DESs and flavonoids, and increases the polarity of the extraction media. Strongly polar chemicals might be extracted more effectively than weakly polar ones using DES with a greater water content [27]. Therefore, DES with more water lowered the effectiveness of extracting flavonoids that were weakly water-soluble. As a consequence, DES-8 was used with a water content of 40% (*v*/*v*).

### 3.4. Effect of Liquid–Solid Ratio on the Extraction Yield of Total Amounts of Flavonoids 

The appropriate liquid–solid ratio is associated with lower cost and solvent consumption in the extract process [26]. According to Figure 2D, the extraction yield of TAFs from mung beans rose from 907.61 ± 92.80 μg/g to 2094.12 ± 20.30 μg/g as the liquid–solid ratio was raised from 10 mL/g to 70 mL/g while decreasing significantly when the ratio further increased to 90 mL/g. The mixture’s density was reduced due to the higher liquid–solid ratio, which also raised the speed at which ultrasonic waves propagated, lessened the influence of ultrasound power attenuation, and improved the transfer of energy over distance and time [28,29], thus leading to a gradual increasing extraction efficiency from 10 mL/g to 70 mL/g. However, the increase in energy consumption of ultrasonication and the reduction in cavitation of ultrasonication might be caused by further increasing the liquid–solid ratio [30]. Therefore, an appropriate liquid–solid ratio is necessary, and 70 mL/g was selected as the optimal ratio.

### 3.5. Effect of Ultrasonic Power on the Extraction Yield of Total Amounts of Flavonoids 

In ultrasonic-assisted extraction processes, ultrasonic power plays an essential function in increasing extraction yields [31]. Herein, UAE-DESs with different ultrasound power (100, 200, 300, 400, 500 W) were utilized to extract mung bean flavonoids. As demonstrated in Figure 2E, the TAFs yields extracted at 200, 300, and 400 W exhibited nonsignificant differences (2174.15 ± 122.07 μg/g, 2219.14 ± 77.65 μg/g, and 2179.65 ± 136.64 μg/g, respectively), but they all were significantly higher than those of 100 and 500 W (1940.66 ± 11.46 μg/g and 2022.55 ± 49.65 μg/g, respectively, *p* < 0.05). The increase in intermolecular vibration caused by the increased ultrasonic power at a suitable range was one potential explanation, which would make it simpler for flavonoids to dissolve in the solvent. Additionally, extraction power affected the size of the cavitation bubbles that were produced during the ultrasonic extraction procedure. The smaller and larger amounts of cavitation bubbles would be created by increasing power, and as they grew and burst, they produced powerful pressure pulses that stirred the material and substrate and increased the output of flavonoids. The energy in the solvent and material, however, decreased when the extraction power was excessive because it might be dispersed to the container wall due to the large number of bubbles that were produced in the solution, and the flavonoid yield would then decrease [32,33]. Furthermore, higher ultrasonic power might also lead to structural damage of flavonoids [31]. Taking into account the principle of energy saving, extraction at ultrasonic power of 200 W should be the optimal choice.

### 3.6. Effect of Extraction Temperature on the Extraction Yield of Total Amounts of Flavonoids 

The bioactive compounds from plants adhere to the matrix through physical adsorption and chemical interactions. Elevated temperature may reduce these interactions, so the secondary metabolites will desorb and dissolve in the extraction solvents more easily [26]. In addition, increased temperature would decrease solvent viscosity, resulting in enhanced mass transfer and diffusion rates. As a result, the extraction process is made more efficient [34]. As depicted in Figure 2F, the TAFs extraction yield initially rose from 1932.86 ± 9.42 μg/g to 2298.23 ± 39.97 μg/g with temperature increasing from 30 °C to 70 °C, and then significantly decreased to 2150.52 ± 31.73 μg/g with further increase in temperature to 80 °C. The solvent viscosity decreased, resulting in an increase in diffusivity and an improvement in the release rate of flavonoids from the sample to the solvent as the ultrasonic temperature increased. However, if the temperature further increases, the heat-sensitive target substance may be degraded, which will result in a lower extraction yield [35]. Therefore, 70 °C was determined as the optimal extraction temperature.

### 3.7. Effect of Extraction Time on the Extraction Yield of Total Amounts of Flavonoids 

Extraction time is an essential variable that influences the extract efficiency [36]. We assessed the relationship between extraction yield and time in the range of 10 to 60 min. The extraction yield did not show significant change (*p* > 0.05), as depicted in Figure 2G, when the extraction duration was increased from 10 to 60 min. Too long of an extraction time will waste energy, so a shorter extraction time of 10 min was considered appropriate.

### 3.8. Model Fitting and Response Surface Methodology

The extremes of liquid–solid ratio, ultrasonic temperature, and water content in DES in the single factor experiment (839, 319, and 312 μg/g, respectively) were the most significant according to the difference between maximum and minimum values, so these three factors were selected to carry out the optimization experiment by RSM. To examine the effects of three independent factors (A, liquid–solid ratio: 60–80 mL/g; B, water content: 30–50%; C, ultrasonic temperature: 60–80 °C) on the response variable (TAFs yield), 17 experiments were conducted in triplicate utilizing the Box–Behnken design (BBD). The levels of the coded independent variables, as well as the experimental and predictive outcomes, are provided in Table 2 and Table 3. Experimental data for the response variables were analyzed using quadratic equations and displayed in the following manner:*Y*(TAFs) = 2144.43 − 153.75*A* + 57.64*B* − 11.33*C* + 100.62*AB* + 22.20*AC* + 29.54*BC* + 15.44*A*^2^ − 115.92*B*^2^ + 25.19*C*^2^(1)

The significance level for the *p*-value is set at 0.05 for the optimized parameters, as stated by Elik et al. [37], in order to evaluate their contribution to the BBD. The ANOVA analysis, as presented in Table 4, revealed that the model was significant (*p* < 0.05) for three response variables. Moreover, the lack of fit, which is not statistically significant (*p* > 0.05), demonstrated that the model effectively depicts the connection between the independent variables and responses [38]. The F-value (4.324) and the *p*-value (0.033) indicated that the suggested model effectively predicted and optimized parameters [29]. The correlation coefficient (*R*^2^) of 0.948 suggested a substantial agreement between the experimental and predicted values [38]. A signal-to-noise ratio above 4.0 indicated a favorable outcome in terms of adequacy precision [19]. The model in our research had an appropriate signal, according to the adequacy precision of 9.190. The outcomes demonstrated that the experimental values for UAE-DES optimized parameters were accurate and reliable. In addition, the F-values and *p*-values of the linear and quadratic coefficients can indicate the extent to which the independent variables affect the response variables [38]. The F-value’s numerical magnitude is directly proportional to its contribution to the BBD [37]. We can see from the results that the F-value of variable A (22.910) is the parameter that contributes the most to the BBD. The linear A coefficient was highly significant (*p* < 0.01) on TAFs’ yield, while the quadratic coefficient B^2^ was significant (*p* < 0.05) on TAFs yield, suggesting that the parameter of liquid–solid ratio (A) and ultrasonic temperature (C) were the two primary factors affecting the extraction yield, and the primary impact relationships of each factor were: liquid–solid ratio > ultrasonic temperature > water content in DES.

The 3D surface plot was commonly used to assess the impact of different variables on the TAFs’ yield. The plot was created by altering two variables within the range of experimental data and maintaining a constant value for the third variable [39]. The extraction yield of TAFs was affected in different ways by various process variables. As the ultrasonic temperature increased, the TAFs in DES showed an upward trend and reached their peak at 70 °C. As the temperature continued to rise to 80 °C, the TAFs slightly decreased (Figure 3A). However, the increase in the liquid–solid ratio led to a reduction in the yield of TAFs. Figure 3B depicts the effect of the liquid–solid ratio and water content in DES on the yield of TAFs at fixed ultrasonic temperature. The yield of TAFs decreased as the liquid–solid ratio increased, whereas water content in DES did not affect the level of TAFs’ yield. At a constant liquid–solid ratio, the TAFs showed an upward trend with the increase in ultrasonic temperature, peaking at 70 °C. However, TAFs showed a slight decline: as the temperature reached 80 °C, the yield of TAFs did not notably change with further increases in water content (Figure 3C). According to the findings in Figure 3 and Equation (1), it was observed that the linear terms of ultrasonic temperature and the liquid–solid ratio had negative impacts on the TAFs extraction. This suggested that as the temperature or liquid–solid ratio increased, the extraction yield of TAFs decreased. The decreased yield could possibly be attributed to the decomposition of flavonoids at high temperatures. Moreover, the moderate positive effect of water content in DES suggested that increasing this factor would improve the extraction efficiency of TAFs. In general, the ultrasonic temperature and the liquid–solid ratio were found to be the major elements that affect the extraction efficiency of TAFs from mung beans by DES-UAE.

### 3.9. Model Validation

Software Design-Expert 8.0 was utilized to optimize the independent and response variables for the suggested extraction. The optimized results were as follows: the liquid–solid ratio (X1) of 60 mL/g, the ultrasonic temperature (X2) of 67 °C, and the water content in DES (X3) of 30%. The predicted highest yield of TAFs was 2383.67 μg/g. To confirm the accuracy of these results, triplicate experiments were examined using the same extraction parameters, and the TAFs extraction yield by DES was 2339.45 ± 42.98 μg/g, which was 1.9 times that by 30% EtOH (1234.75 ± 101.56 μg/g). The chromatogram of the flavonoid profile of sample extracted under the optimal condition was shown in Appendix A. The outcome demonstrated the dependability and appropriateness of the model.

### 3.10. Antioxidant Capacities of the Flavonoids

It is reported that flavonoids exhibit excellent antioxidant ability and they play an important role in human health [7]. An earlier research discovered that the mung bean was rich in flavonoids, which showed DPPH and ABTS free radical-scavenging activity [40]. In this study, the DPPH· and ABTS· scavenging activities of TAFs extracted either by the optimal DES or 30% EtOH were compared, with VC as a contrast.

#### 3.10.1. 2,2-Diphenyl-1-picrylhydrazyl Radical-Scavenging Activity

The DPPH· is commonly utilized to assess the antioxidant ability to scavenge free radicals. This stable free radical has gained significant popularity for this purpose [41]. In the DPPH· assay, the antioxidants were capable of reducing the stable DPPH· to form the yellow-colored diphenylpricryhydrazine. It is the hydrogen-donating ability that contributes to the effect of antioxidants on DPPH radical scavenging [39]. As depicted in Figure 4A, a correlation was observed between the concentration of all samples and their ability to scavenge DPPH·. More specifically, when the concentration of DES extract increased to 30 μg/mL, the DPPH· scavenging activity increased to 90.86% ± 2.37%, which was lower than VC (95.71% ± 2.43%) but higher than 30% EtOH extract (87.14% ± 2.85%). Moreover, the DPPH· scavenging activity also exhibited a significant difference (*p* < 0.05) between DES extract and 30% EtOH extract. Based on these findings, we can infer that the flavonoids extracted from mung bean using UAE-DES under optimal conditions exhibited higher DPPH· scavenging activity compared to that of 30% EtOH.

#### 3.10.2. 2,2′-Azinobis-(3-ethylbenzthiazoline-6-sulphonate) Radical-Scavenging Activity

The ABTS free radical cations are highly soluble in water and alcohol, making them commonly utilized for assessing the antioxidant activity of water-soluble and fat-soluble substances [42]. As shown in Figure 4B, ABTS· scavenging activity of all samples increased with increasing concentration. VC exhibited the highest radical scavenging activity at 30 μg/mL (94.37% ± 4.49%), followed by DES extract (90.14% ± 4.59%) and 30% EtOH extract (87.32% ± 4.73%) with significant differences (*p* < 0.05). We can draw a conclusion that the DES extract exhibited better ABTS· scavenging capacity than the 30% EtOH extract.

### 3.11. Scanning Electron Microscope

To explore the reason why a difference exists in the extraction rate between DES and 30% EtOH, the raw mung beans and those treated by DES or 30% EtOH were examined by SEM, and their micrographs are shown in Figure 5. There were obvious morphological changes on the surface of the mung bean caused by all extraction solvents (Figure 5C–F) when compared to the untreated sample (Figure 5A,B). The surface tissue of the raw mung beans was smooth and compact (Figure 5A,B). It was found that the surface of the mung bean showed more visible pores and chasms after extracted by UAE with 30% EtOH (Figure 5C,D). However, after being extracted by DES, the surface of the mung bean was rougher than after being extracted by 30% EtOH, with many irregular shapes on it as if the surface had been corroded (Figure 5E,F). This might be attributed to the destruction of the cell wall, which is caused by the ability of DES to dissolve and/or hydrolyze cellulose. As a result, the solvent extraction method exposed a greater quantity of target compounds compared to the 30% EtOH extraction.

### 3.12. Fourier Transform Infrared Spectrometer

The FTIR spectra were conducted to research the intermolecular interactions between flavonoids and DES, including DES, vitexin monomer, isovitexin monomer, vitexin-DES, and isovitexin-DES samples. As can be seen in Figure 6, taking isovitexin, for example, there were C=O variation and C-OH absorption bands in isovitexin monomer and isovitexin-DES samples. In addition, the blue shifts were found in the stretching and deformation vibrational absorption bands of C-OH, which shifted from 1180.60 to 1031.34 cm^−1^ and 1353.73 to 1204.48 cm^−1^, respectively. These results proved that a new hydrogen bond was generated between isovitexin and DES and also demonstrated that there was a difference in conformation between isovitexin-DES and isovitexin monomer. The similarity between the earlier research and our research is that both observed shifts in the stretching and deformation vibration absorption bands of ellagic acid’s C-OH group, which shifted from 1193.09 to 1188.77 cm^−1^ and 1331.55 to 1313.36 cm^−1^, respectively [33]. Additionally, there was an increase in the absorption band of C=O in isovitexin from 1649.25 to 1734.33 cm^−1^, suggesting the presence of a hydrogen bond between DES and the carbonyl group of isovitexin. The O-H stretching vibration absorption band of the free hydrogen bond in isovitexin was observed at 3382.08 cm^−1^. However, when isovitexin was dissolved in DES, this absorption band shifted to 3305.97 cm^−1^. It is possible that the free OH group in isovitexin formed an intermolecular hydrogen bond with DES, which led to the observed shift in the absorption band [33]. In summary, the hydrogen bonding between DES and flavonoids enabled a higher extraction yield.

## 4. Conclusions

In our study, a green DES solvent was introduced to evaluate the extract efficiency of mung bean flavonoids. We examined the elements that affect DES-UAE technology and further optimized them. Among the different types of DESs examined, ethylene glycol-glycolic acid was selected as the most promising DES solvent. The single-factor and RSM experiments were conducted to optimize the extraction conditions, yielding the following optimal results: liquid–solid ratio of 60 mL/g, water content in DES of 30%, ultrasonic power of 200 W, ultrasonic temperature of 67 °C, and ultrasonic time of 10 min. The TAFs extraction content was 2339.45 ± 42.98 μg/g under these optimal parameters, which was significantly higher than that of the UAE-30% EtOH extract (1234.75 ± 101.56 μg/g, *p* < 0.05). Furthermore, the DES extract exhibited better antioxidant activity (DPPH· and ABTS· scavenging activity) than the 30% EtOH extract. SEM results showed that there were significant microscopic morphology changes in the surface structure of the cell wall caused by UAE-DES extraction. FT-IR results indicated that an intermolecular hydrogen bond formed between DES and mung bean flavonoids. The DES-UAE technology offers a promising and eco-friendly method for flavonoid extraction, boasting efficiency, time-saving, and environmental benefits over traditional solvents, aligning with sustainability goals. Despite its potential, further research is needed to refine parameters, assess quality, and address challenges in scaling up production and purification processes.

## Figures and Tables

**Figure 1 foods-13-00777-f001:**
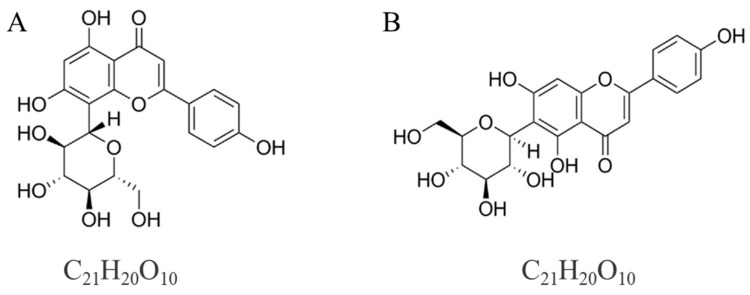
Chemical structures of (**A**) vitexin and (**B**) isovitexin.

**Figure 2 foods-13-00777-f002:**
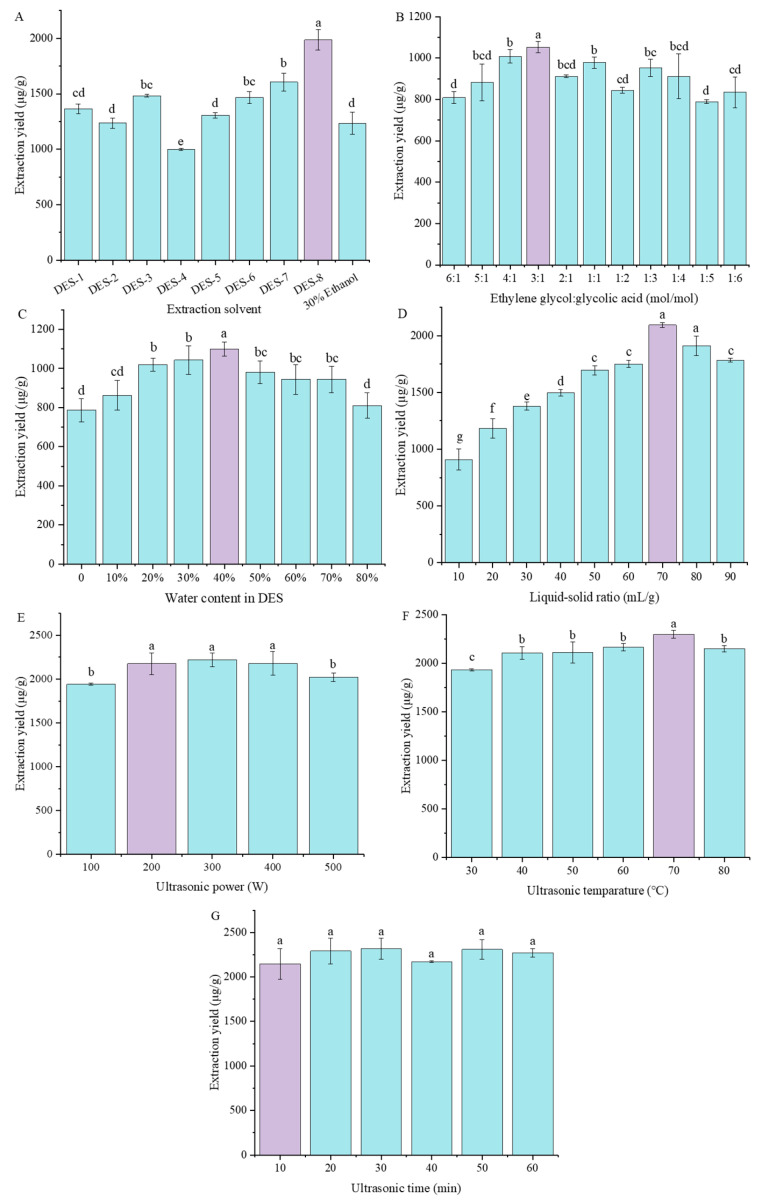
Effects of extraction solvent (**A**), DES molar ratios (**B**), water content in DES (**C**), solid-to-liquid ratio (**D**), ultrasonic power (**E**), ultrasonic temperature (**F**), and ultrasonic time (**G**) on extraction yields of total amounts of flavonoids from mung beans within single factor experiments. Different letters indicate significant differences (*p* < 0.05). Purple in each figure represents the optimal condition of each parameter. DES-1: choline chloride-urea; DES-2: choline chloride-1,2-propanediol; DES-3: choline chloride-ethylene glycol; DES-4: choline chloride-citric acid; DES-5: choline chloride-malic acid; DES-6: ethylene glycol-malonate; DES-7: 1,2-propanediol-glycolic acid; DES-8: ethylene glycol-glycolic acid. The molar ratio between component 1 and 2 of all solvents was 1:2. The optional paraments were as follows: DES-8 solvent (**A**), molar ratio between ethylene glycol and glycolic acid of 3:1 (**B**), water content in DES of 40% (**C**), liquid–solid ratio of 70 mL/g (**D**), ultrasonic power of 200 W (**E**), ultrasonic temperature of 70 °C (**F**), extraction time of 10 min (**G**).

**Figure 3 foods-13-00777-f003:**
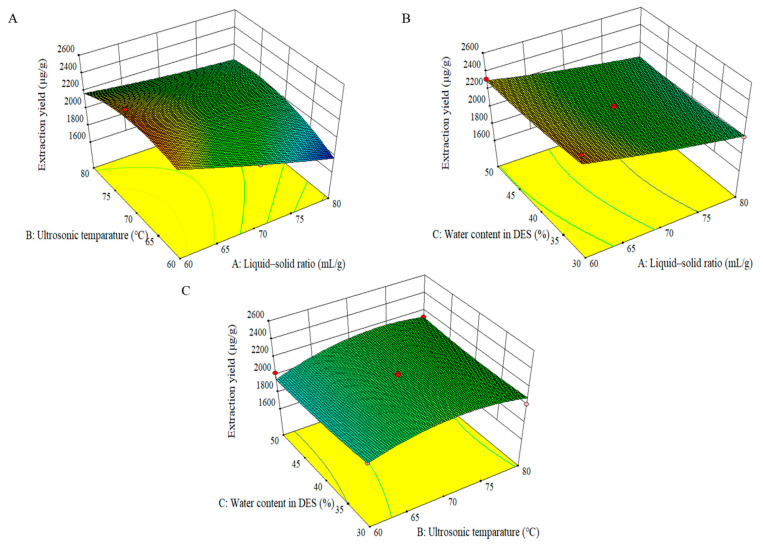
3D response surface plots showing the interactive effects of the independent variables on the total amounts of flavonoid yield (μg/g). Liquid–solid ratio and ultrasonic temperature (**A**); Liquid–solid ratio and water content in DES (**B**); Ultrasonic temperature and water content in DES (**C**).

**Figure 4 foods-13-00777-f004:**
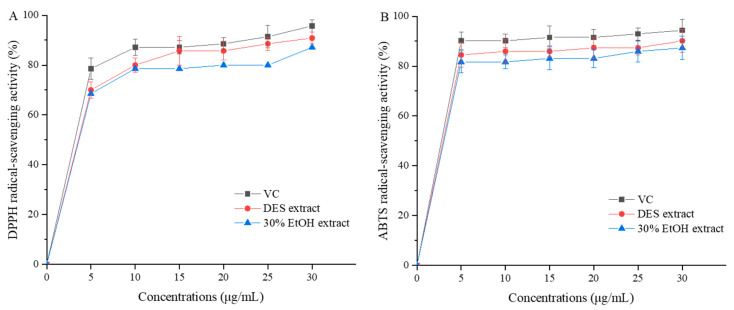
DPPH radical-scavenging activity (**A**) and ABTS radical-scavenging activity (**B**) of the flavonoids extracted from mung bean by DES (ethylene glycol-glycolic acid) and 30% ethanol with VC as a positive control.

**Figure 5 foods-13-00777-f005:**
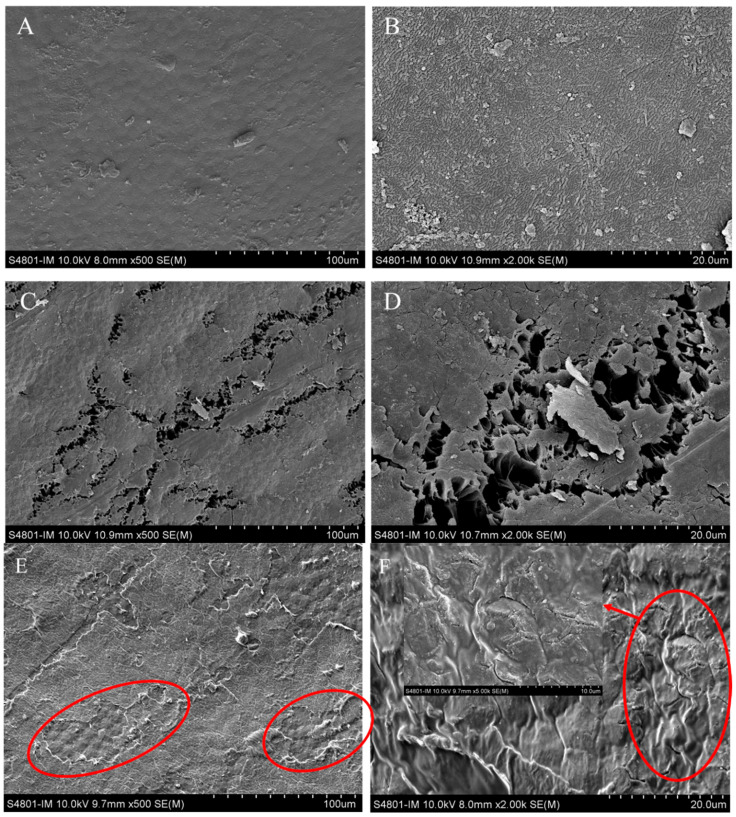
Scanning electron microscopy of original mung bean (**A**,**B**), mung bean after extraction by 30% ethanol (**C**,**D**), and mung bean after extraction by DES (ethylene glycol-glycolic acid) (**E**,**F**). The red circular area denotes the exchange in mung bean after extraction by DES.

**Figure 6 foods-13-00777-f006:**
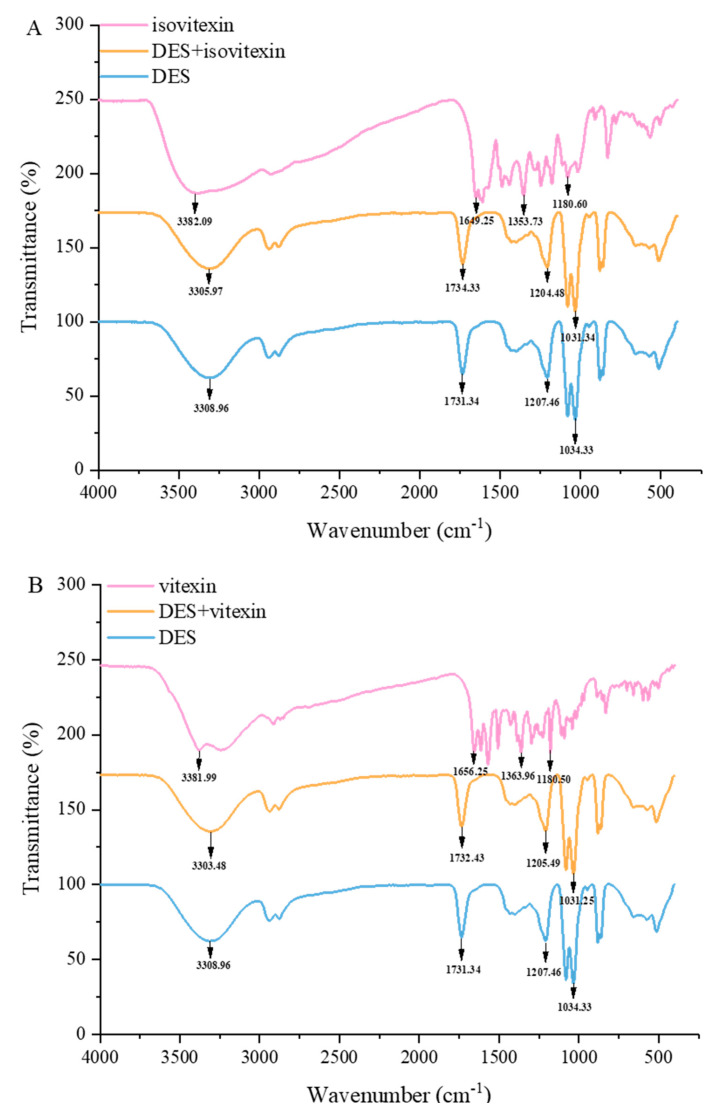
Fourier transform infrared (FT-IR) spectra of isovitexin, DES-8, and DES-8+isovitexin (**A**); Fourier transform infrared (FT-IR) spectra of vitexin, DES-8, and DES-8+vitexin (**B**).

**Table 1 foods-13-00777-t001:** Different types of deep eutectic solvent (DES) selected.

DES Groups	Component 1	Component 2	Molar Ratio
DES-1	choline chloride	urea	1:2
DES-2	choline chloride	1,2-propanediol	1:2
DES-3	choline chloride	ethylene glycol	1:2
DES-4	choline chloride	citric acid	1:2
DES-5	choline chloride	malic acid	1:2
DES-6	ethylene glycol	malonate	1:2
DES-7	1,2-propanediol	glycolic acid	1:2
DES-8	ethylene glycol	glycolic acid	1:2

**Table 2 foods-13-00777-t002:** Independent variables and levels used for Box–Behnken design (BBD).

Factors	Level		
	−1	0	1
Liquid–solid ratio (X1) (mL/g)	60	70	80
Ultrasonic temperature (X2) (°C)	60	70	80
Water content in DES (X3) (%)	30	40	50

**Table 3 foods-13-00777-t003:** Box–Behnken design (BBD) matrix and response values for the extraction yield of total flavonoids from mung beans.

Std.	X1	X2	X3	Extraction Yield (μg/g)
1	0	1	−1	2018.97
2	1	0	1	1951.25
3	0	−1	1	2029.34
4	0	0	0	2175.33
5	0	0	0	2139.67
6	−1	0	−1	2463.26
7	0	−1	−1	2034.37
8	0	0	0	2045.17
9	−1	−1	0	2152.30
10	1	0	−1	2006.24
11	−1	0	1	2319.49
12	1	−1	0	1748.71
13	0	1	1	2132.12
14	1	1	0	2136.82
15	−1	1	0	2137.94
16	0	0	0	2189.57
17	0	0	0	2172.41

**Table 4 foods-13-00777-t004:** Analysis of the variance (ANOVA) for the second-order polynomial model.

Source	Sum of Squares	df	Mean Square	*F*-Value	*p*-Value
Model	321,245.700	9	35,693.960	4.324	0.033 *
A-Liquid–solid ratio	189,102.300	1	189,102.300	22.910	0.002 **
B-Ultrasonic temperature	26,581.520	1	26,581.520	3.220	0.116
C-Water content	1026.892	1	1026.892	0.124	0.735
AB	40,495.180	1	40,495.180	4.906	0.062
AC	1970.830	1	1970.830	0.239	0.640
BC	3491.468	1	3491.468	0.423	0.536
A^2^	1003.442	1	1003.442	0.122	0.738
B^2^	56,581.020	1	56,581.02	6.855	0.035
C^2^	2671.865	1	2671.865	0.324	0.587
Residual	57,778.800	7	8254.114		
Lack of fit	44,128.860	3	14,709.620	4.310	0.096
Pure error	13,649.940	4	3412.484		
Total	379,024.500	16			
R-Squared = 0.948	Std.Dev. = 90.852				
Adjusted R-Squared = 0.852	Mean = 2109				
Predicted R-Squared = 0.919	C.V. % = 4.310				
Adequacy precision = 9.190					

Note: Symbols * and ** represent the significant differences at *p* < 0.05 and *p* < 0.01, respectively.

## Data Availability

The original contributions presented in the study are included in the article/Appendix A, further inquiries can be directed to the corresponding author.

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
