# Peer review of "Deep Eutectic Solvents as New Extraction Media for Flavonoids in Mung Bean"

_foods, 2024, doi:10.3390/foods13050777_

Round 1

Reviewer 1 Report

Comments and Suggestions for Authors

The paper contains all the requirements of a quality scientific article. The methods used are appropriate and sufficiently described so that the results are accurate and easy to interpret.

The graphics are very suggestive and correctly interpreted. The paper has served its purpose. I congratulate the authors for their work and effort.

Reviewer 2 Report

Comments and Suggestions for Authors

Dear, the manuscript "Deep Eutectic Solvents as a New Extraction Media for Flavonoids in Mung Bean" is quite interesting and worth investigation. Please see some comments below:

1- Have you considered water,only?

2- Some people consider ultrasonic thank too weak. Thus, it should used to clean glassware, only. However, it preserve most of all compounds, chemically. 

3- Have you considered a kinects? It is fundamental.

4- Please add the cromatograms (supplementary material)

5 -  HPLC does not detect conjugated flavonoids? The extraction method may act on this.

6- I do not agree with you experimental design. You used a water% higher than you preliminary study. It should be at least with internaval

7- Did you validated the optimal condition?

8- You have HPLC, antioxidant methods are not needed. Please focused on HPLC data. The other can be superficially discussed. The same to FT-TR

Regards

Reviewer 3 Report

Comments and Suggestions for Authors

This is a very good piece of work; it is usefu and interesting.

However, there are some weak points that authors would need to improve. 

One of them, and perhaps the weakest one, is the origin of samples. They were bought in a market and in this way there is no information on the characteristics and origin of samples. In this way, reproducibility of results byh other potential researchers is rather weak. 

References- There are two to three recent studies, closely related to this study, which were not taken into consideration by authors.

Results - By leaving outside key published studies the full analysis of results has weak aspects that inhibit a better understanding of the whole work. However, in spite of these comments, this research has several and good results for understanding the objectives provided in the design of this research work.

Conclusions - They have for sure a weak component that deserves to be improved. Based on the results obtained in this study, where future research studies should be addressed to ? To which basic and outstanding components should be taken into consideration in future studies, but based on the outstanding results provided for this useful work.

Authors are invited to take into consideration the items described here in order to improve the quality of this interestng report. And I would, finally, invite authors that in future research works, they should use raw materials on which the origin and characteristics may be well identified.

Reviewer 4 Report

Comments and Suggestions for Authors

Dear Authors,

The manuscript titled "Deep Eutectic Solvents as a New Extraction Media for Flavonoids in Mung Bean" investigates the use of deep eutectic solvents (DES) combined with ultrasound-assisted extraction (DES-UAE) for extracting flavonoids from mung beans. It compares the efficacy of eight different DESs against traditional 30% ethanol extraction, with the finding that ethylene glycol-glycolic acid DES yielded the highest flavonoid extraction, outperforming the traditional method. The optimized parameters for DES-UAE resulted in a significant increase in extraction yield and antioxidant activities compared to the ethanol extract. The study concludes that DES-UAE is an efficient, sustainable method for extracting high-quality flavonoids from mung beans.

Deficiencies in Abstract

Lack of context about why flavonoid extraction is significant or the implications of these findings.

It does not address any limitations or future directions for research stemming from the study's findings.

Deficiencies in introduction 

The introduction outlines the importance of mung beans in food and medicine, noting their bioactive compounds and health benefits. It discusses the focus on flavonoids due to their antioxidant properties and the traditional use of organic solvents for extraction, which poses environmental concerns. The manuscript highlights the innovative use of deep eutectic solvents (DES) as a greener alternative, aiming to develop an effective DES method assisted by ultrasound to extract flavonoids and evaluate its efficiency and environmental impact.

Deficiencies in MM

The Materials and Methods section details the collection, preparation of mung beans, and the formulation of various deep eutectic solvents (DESs) for flavonoid extraction. It describes the ultrasound-assisted extraction process, optimization experiments, and analytical methods including HPLC, SEM, FT-IR, and antioxidant capacity assessment. Despite thoroughness, it could benefit from clarifying the selection criteria for beans and DES components, the rationale behind specific extraction parameters, and more immediate accessibility to procedural details currently relegated to supplementary materials. Enhanced detail would improve reproducibility and understanding.

The Results and Discussion section effectively presents the findings on flavonoid extraction from mung beans using DES, comparing it to traditional ethanol extraction, and investigates the impact of various parameters on yield and antioxidant capacity. However, potential areas for improvement include:

Comparative Analysis: A more detailed comparison with existing studies could contextualize the findings within the broader research landscape.

Mechanistic Insights: While the section touches on the extraction mechanism via SEM and FT-IR analysis, deeper insights into the molecular interactions between DES components and flavonoids could enrich the discussion.

Limitations and Future Work: Acknowledging the study's limitations and proposing specific future research directions could enhance the manuscript's contribution to the field.

Environmental Impact Discussion: An expanded discussion on the environmental benefits of using DES over traditional solvents, including any potential drawbacks, would provide a more comprehensive assessment of the method's sustainability.

For conclusion

Broader Implications: Elaborating on the broader environmental and health implications of adopting green extraction technologies.

Comparative Advantages: More explicitly stating the advantages of DES-UAE over traditional methods beyond yield and antioxidant activity, considering factors like cost, scalability, and environmental impact.

Future Directions: Offering clearer insights into how this research could influence future studies or the development of green technologies in biochemistry and pharmaceuticals.

Final decision and revision suggestions would typically focus on enhancing the manuscript's clarity, ensuring the methodology is thoroughly explained, results are accurately presented, discussions align with findings, and conclusions are supported by data. 

Comments on the Quality of English Language

Minor editing of English language required.

Round 2

Reviewer 2 Report

Comments and Suggestions for Authors

The authors have addressed almost all comments properly. In fact, I think the kinects is fundamental to get a clear trend on it. However, the currently version is publishable.

Reviewer 4 Report

Comments and Suggestions for Authors

Accept.

Comments on the Quality of English Language

Minor editing of English language required.